# An Approach to the Semantic Representation of the Local Government Strategic Planning Process: Ontology-Driven Simulation Method for Assessing Economic Impacts

Miroslav Zarić [1], Saša Arsovski [2], Branko Markoski [1,*], Petar Vasiljević [3] and Velibor Premčevski [4]

1 Faculty of Technical Science, University of Novi Sad, 21000 Novi Sad, Serbia
2 Faculty of AI and Robotics, Raffles University, Johor Bahru 79250, Johor, Malaysia
3 Centre for Information Technologies (CIT-UNS), University of Novi Sad, 21000 Novi Sad, Serbia
4 Technical Faculty "Mihajlo Pupin", University of Novi Sad, 23000 Zrenjanin, Serbia
* Correspondence: markoni@uns.ac.rs

**Abstract:** This paper describes a methodological approach to the creation of a strategic planning ontology (SPO) for local governments. The aim of the given ontology is to provide a semantic description of the development strategy, involved stakeholders, and tools to improve the decision-making process for selecting the most valuable strategic objectives for the economy of the region. The novelty of the research presented in this paper is reflected by the development of an ontological model of actions and reasons, that is, semantically described combinations of activities and requirements that are defined by development strategies. The proposed ontology can provide answers to questions such as: what are the development priorities? How will those priorities be implemented? Who will be responsible for implementation? Why are they being implemented in the first place? We answer the question, "why specific development goals are chosen?". Answers are given as semantic representations of the economic impact on a region's industry. The ontology-driven simulation methods that are proposed in this paper provide the possibility of simulating the effects and results of the chosen development strategy goals, evaluate the actions taken, and ensure support for the selection of the best alternative in the process of defining objectives for the development strategy.

**Keywords:** cooperative systems; decision support systems; intelligent systems; local government; ontologies; strategic planning

## 1. Introduction

This paper is an extended version of the paper [1] published in the international conference: 6th International Conference on Information Society and Technology, Kopaonik, Serbia, 2016.

A strategic plan is a document that contains several strategic goals. It provides guidelines for domain development activities over a period of three to five years and determines the direction, priorities, actions, and responsibilities of implementation.

Standard developmental priorities and goals of state and local administrative bodies are [2]: *Smart growth*—the development of a knowledge and innovation-based economy. *Sustainable growth*—promoting a more efficient economy that makes better use of available resources. *Inclusive growth*—promoting the development of sectors with high employment rates and accomplishing social and territorial cohesion.

The strategic planning (SP) process is defined as the process by which managers of companies analyze internal and external environments to formulate the strategy and the allocation of resources to develop a competitive advantage in the industry that enables the achievement of the organization's objectives [3]. From the above definitions and the new needs of the organization, two key issues of formalization should be addressed and resolved in the context of strategic planning [4]:

(1) To define a conceptual framework for representing information/knowledge extracted from the internal and external environment of the organization.

(2) To determine steps in the strategic planning process, type of information/knowledge, and formalize the strategic planning process itself.

To allow a higher level of automation and formalization of the strategic planning process, the following tools and software have been utilized: competitive intelligence tools (CI) [5] and business intelligence (BI) [6]. These tools have not been integrated into all phases involved in strategic planning because they only provide reports and documents that decrease the agility of the subsequent revisions of the Strategic Plan [7].

Guarino [8] proposes ontology classification by the level of generalization and discusses four different types: upper, domain, task, and application ontologies. Most ontologies created for planning purposes are designed for specific domains: budget planning [9], public healthcare projects [4], public transport organizations, and action planning [10]. Guizzardi [11] discusses that formal ontology, as a discipline, and representation languages, based on formal ontological principles, are essential for achieving interoperability as one of the means of maximizing the added value of information artifacts.

Methodologies used to develop ontologies specifically aimed at planning have been insufficiently researched. The continuous development and transformation of knowledge are the main reasons for the relatively small number of different methodologies for creating planning-oriented ontologies. The main advantage of using ontologies to describe knowledge of a continuously changing domain is their ability to develop and transform knowledge into a set of semantic descriptions and rules [12].

A schematic description of the strategic planning process can be found in Walsh [13]. That description defines several steps in strategic planning workflow that correspond with answering fundamental questions:

1. Where we are now?—Understanding current state—starting position;
2. What result do we want?—Defining goals;
3. What deliverables (products/services) are provided?—Defining objectives;
4. How are we going to do it?—Defining the action plan;
5. How are we doing? Are we meeting goals?—Specifying monitoring activities;
6. Where do we want to be a year x?—Defining future vision.

Analyzing the proposed description of the strategic planning workflow, the answer to the question "where are we now?" was obtained by the SWOT analysis of the observed region. The next steps in the SP process are to decide on the objectives and results to be achieved by applying a strategy. According to [6], the strategic objectives are created in tandem with the statement of vision and provide an answer to the question "where do we want to be in the future?". However, the vision of the strategy created in this way does not answer the question of *why* we select specific strategic goals. The adoption of vision statements and the definition of the objectives of the strategy is a decision-making process.

Sage [14] analyzed systems and processes for planning and decision support. The authors' research supports the design of information systems capable of coping with the needs and fundamental limits of improved judgment. Sage [14] discussed and described the following: (1) the cognitive styles of decision makers; (2) individual human information processing in decision situations and biases in the acquisition, analysis, and interpretation of information; (3) decision rules for individual decision situations; (4) contingency task structural models of decision situations; and (5) decision-making frameworks, organizational settings, and information processing in group and organizational decision situations. Sutherland [15] stated that the success and survival of an organization depend on its ability to adapt to specific contingencies. This has led to investment in systems for information management and decisions in the planning function. Kainuma et al. [16] proposed an integrated decision support system for use in environmental planning. The purpose of developing an integrated decision support system is to systematically support a series of tasks from systems analysis to policy analysis with the aid of these systems science

methodologies and by integrating the knowledge or judgment of experts with the available numerical data.

According to the rational model of decision-making, the decision-making process can be divided into six steps, as described in [17]:

1. Identifying the problem;
2. Generating alternatives;
3. Evaluating an alternative;
4. Choosing an alternative;
5. Implementing a decision;
6. Evaluating decision effectiveness.

After identifying the problem, alternatives for its solving should be generated. These alternatives are evaluated, and the best alternative is selected for implementation. The selected alternative is then assessed over time to ensure the effectiveness of the decisions. After this assessment, the whole process may be restarted to achieve better outcomes.

In this study, we suggest a novel approach to defining strategic goals and objectives. Our approach is based on the need to integrate business intelligence tools into a strategic planning process. For this purpose, we propose an ontology-driven simulation method for assessing the economic impacts that should be implemented as a decision support tool for choosing the best strategy goals and objectives.

The proposed ontological model semantically describes the strategic planning process and is used as a basis for ontology-driven simulation methods for assessing the economic impact of the chosen strategy.

Machado [18] presents three common dichotomies: reference vs. application ontologies, description logics vs. resource description framework (RDF)-based semantics, and ontological vs. conceptual models.

Ontology is annotated with a semantic representation of input–output multipliers obtained from the economic impact analysis of the analyzed region. Economic impact analysis provides a rule-based and transparent measure of economic activity, jobs, wages, and tax revenues [5]. According to the author of [5], the results of the economic impact analysis can be seen through the increase/decrease of the following indicators:

1. The operating result (or sales volume);
2. Additional value (or gross regional product);
3. Wealth (including property values);
4. The personal income, tax (salary);
5. New jobs.

Improvement of the above-listed indicators is the main objective of most development strategies.

Economic impacts can be classified as follows [5]:

1. Direct economic impacts—changes in the economy of the observed region that occur as a direct result of public or private business decisions and strategic public policy.
2. Indirect economic impacts—growth or decline in business (suppliers, including trade, services, etc.) due to the consequences of changes in the operations of companies that are affected by direct economic impacts.
3. Induced economic impacts—economic effects that are reflected in the increase/decrease in spending on food, clothing, and consumer services, occurring as a result of changes in the income of workers in companies that are directly or indirectly affected by development policies.
4. Dynamic economic impacts—seen through changes in the composition of the population over time, changes in business locations, or methods of land use.

The discussion above makes it clear that strategic planning is a decision-making process, based on multiple, and sometimes opposing criteria. Multicriteria decision-making (MCDM) techniques assist decision-makers in dealing with large amounts of complex

information. Some approaches to MCDM are discussed in brief in the next section in correlation with the proposed ontology.

This paper is organized as follows: The Section 2 provides an overview of the related work. The Section 3 presents the ontology conceptualization process. The Section 4 presents an ontology annotation process. The Section 5 shows the prototype use of the SPO ontology designed to semantically represent the SP process and decision support in this process. In the Section 6, concluding remarks and directions for further research are provided.

## 2. Related Work

The authors in [7] defined the conditions that need to be satisfied to create a good development strategy:

1. It needs to be founded on a clear understanding of the current situation and the need for development;
2. It needs to be relevant and clearly defined;
3. It involves both people and institutions;
4. It is logically consistent;
5. It is realistic and applicable;
6. It needs to have a real influence on development processes;
7. It enables monitoring and creates the foundations of responsibility and transparency whilst being susceptible to change;
8. It is broad enough to encompass all main development problems;
9. Authenticity.

The strategic planning process has the following phases of development, as discussed in [19]:

1. Preparation

The mission and strategic axes were determined during this phase. Strategic axes represent the most general strategic orientation. Strategic axes are sets of decisions that integrate various activities and strategic plans in response to contextual changes in the medium or long term. Some examples of strategic axes are: *reforming the institutions and modernizing the administration*—devoted to strengthening governance in its political, administrative, security, economic, and local dimensions; *developing human capital*—emphasis on education, vocational and technical training, health, access to drinking water and sanitation; or *stimulating sectors with growth potential for the economy and employment*—refers to the improvement of agricultural productivity or increase in the added value of industries and service industries.

2. Collection and analysis of information

The best way to ensure that the analysis encompasses all questions emerging from data is the use of SWOT analysis (strengths, weaknesses, opportunities, threats).

3. Development of a strategy (visions and goals of the action plan)

Strategies need to represent a sequence of goals that grow progressively in more detail and develop a detailed action plan. There are three goals: strategic goals, priorities, and specific goals. The model of strategic goals, priorities, and specific goals, and, finally, the action plan need to be logical and hierarchical. Higher goals need to logically lead to lower goals, and lower goals need to contribute to the accomplishment of higher goals. Priorities and specific goals in the framework of every strategic goal need to support and supplement priorities within these strategic goals.

4. Finish (conclusion and implementation of the strategy)

Bolton [19] defined the requirements that every development strategy needs to fulfill:

(a) Consistency—All strategic goals, priorities, and specific goals must be in mutual accordance.
(b) Relevance—Goals must be relevant to the given situation.

(c) Specifiability—Priorities and goals need to be specific enough to ascertain whether they have been accomplished.

The authors of [10] define the principles of designing planning-oriented ontologies:

- Ontologies need to have a well-defined goal and support a defined group of usage cases;
- Ontologies need to have a minimal number of different concepts and traits;
- Ontologies need to be higher than simple domain concept taxonomies;
- They need to enable the import/export of concepts and traits from other ontologies.

The goal of the ontology described in [5] is to describe plans that can be used by different tools and applications. The authors of [7] defined the strategic planning (SP) process as an analysis of internal and external factors with the goal of forming strategies and allocating resources. They have also presented the ontology formalized by Ontology Web Language (OWL), which meets the standards approved by the World Wide Web Consortium (W3C) and its use for the formalization of the SP process, as well as the knowledge that is created and flows between the participants in the process. According to the authors of [5], the strategic planning ontology design process, based on the model proposed in [3], includes the following:

1. Concept formalization involves all concepts in the strategic planning process, their properties, and relationships between them.
2. The formalization of procedures to perform them in the SP process, obtained from the relationships between the various concepts.
3. Formalization of stakeholders associated with a particular task or concept in the SP process.

The authors of [5] define concepts represented with a hierarchical structure that determines the inheritance of properties between a parent node (or concept) and the child node (or concept). All concepts directly or indirectly inherit the properties of an object. The IS-A relation is a relation that organizes terms or concepts from general to specific ones. The hierarchical structure of concepts is represented by the IS-A relation. Specific inherent features of the general IS-A relation are the backbone of knowledge organization and sharing [10].

The authors of [17] define three steps of the SP process:

1. The first step involves creating an instance of the concept mission. The description property of the created instance contains a value that defines the mission of the organization in the SP process.
2. In the second step, instances of concepts are created. These instances define the types of information and stakeholders.
3. In the third step, ontology is used to integrate information and knowledge into the SP process.

According to the authors of [7], the ontological structure (explicit specification of the conceptualization of the domain) can be formally defined as follows: $O = \{C, R, A^o\}$, where $C$ is a set of elements called concepts/classes, with properties describing their features and attributes. $R \subseteq C \times C$ is the set of relationships between the concepts/classes of $C$, which is defined so that it represents hierarchical taxonomy. $A^o$ is the set of axioms in $O$ that impose restrictions on the concepts and their relationships. The lexicon (language) $L$ represents common vocabulary regarding conceptualization $O$ and is defined as $L = \{L^C, L^R, F, G\}$, where $L^C$ is the set of elements called lexical entries of concepts, $L^R$ is a set of elements called the lexical entries of relationships. $F \subseteq L^C \times C$ is a reference to concepts that establish the link between a concept and a lexical entry. $G \subseteq L^R \times R$ is a reference to relationships that establish the link between a relationship and a lexical entry.

Using this previous definition, the authors of [7] have then formally defined ontology as $O_m = <O, L>$, where $O$ is the ontological structure and $L$ is the corresponding associated lexicon (language). In the same paper, the authors presented a detailed hierarchical view of ontology for the SP process.

Different methodologies are used to predict the economic impact of development strategies on individual income, tax, employment, business, sales, and value-added of the observed region.

To identify multipliers that represent direct, indirect, induced, and dynamic types of economic impacts, the authors propose the use of an input–output model (I/O). The proposed model identifies links between industry, purchase, and sale of the observed environment (region, country) [19]. The input–output model is a quantitative economic technique that represents the interdependencies between different branches of the national economy or different regional economies [20].

Wassily Leontief (1905–1999) is credited with the development of this model [21]. Leontief won the Nobel Prize in Economic Science for the development of this model.

The model states that sectors in the economy are dependent on other sectors because, to produce something, each sector needs to consume output from other sectors. In an economy with I sectors, each sector produces x units of a single homogeneous good. Each sector sells some of its output to other sectors and some of its output to consumers (final output x, final demand c). In inter-industry matrix A, column entries typically represent inputs to an industrial sector, while row entries represent outputs from a given sector. Therefore, the matrix is often referred to as the consumption matrix (Equations (1) and (2)).

The coefficients in row one correspond to industry 1, and the coefficients in row two are the amounts necessary for industry 2 to produce something. Therefore, this format shows how dependent each sector is on every other sector, both as a customer of outputs from other sectors and as a supplier of inputs [22]. A mathematical representation of the I/O model is given in Equation (1):

$$x = \begin{vmatrix} x_1 \\ . \\ . \\ . \\ x_n \end{vmatrix}; \; c = \begin{vmatrix} c_1 \\ . \\ . \\ . \\ c_n \end{vmatrix}; \; I = \begin{vmatrix} 1 & 0 & 0 & 0 \\ 0 & 1 & 0 & 0 \\ 0 & 0 & 1 & 0 \\ 0 & 0 & 0 & 1 \end{vmatrix}; \tag{1}$$

$$A = \begin{vmatrix} a_{11} & . & . & a_{1n} \\ . & . & . & . \\ . & . & . & . \\ a_{n1} & . & . & a_{nn} \end{vmatrix}; \tag{2}$$

$$Ax + c = x; \; (I - A)x = c; \; x = (I - A)^{-1}c \tag{3}$$

Equation (1) Mathematical representation of the I/O model.

According to the author of [19], economic simulation models such as the I/O model can be used directly in the assessment of the effects on income and employment by changing the level of business activity (e.g., industry closings or openings), and with "demand translator" (budget breakdown) tools, to assess the effects of changes in spending (for example, tourism or construction activities). However, I/O models are flawed because they do not cover dynamic influences over time. Simulation models, such as I/O models, can be directly applied to assess the effects of changes in income induced by changes in business location and/or industrial activities.

The authors of [23] described the multipliers generated by the regional (I/O) model. Regional I/O multipliers measure the goods and products of each industry and the use of these goods from other industries and end users. Regional I/O multipliers provide information on inter-industrial relations and changes in demand in the industrial sector of the region. The input–output model (I/O) of economics uses a matrix representation of a region's economy.

To assess the economic impact, the authors of the Australian Institute for Social Research recommend the RISE v3.0 model [24]. Rise v3.0 is a model created in Microsoft Excel, designed to help regional analysts understand the structure of the regional economy and assess the economic impact of changes in the region. The RISE v3.0 model is a

standard input–output (I/O) model expanded to measure the impact of growth or decline in employment on the population of the region.

The model has three functions:

1.　Describe the industry structure;
2.　Measure regional economic impact;
3.　Provide an input–output table and multipliers for the region.

The model can be used to assess the impact of final consumption on the economy of the region over a period of ten years. The model can simulate the launch of new industries and their impact on the region.

As well as ontology-driven models for decision-making problems, different MCDM techniques have been study topics since the 1960s. All decision-making problems, as is the case with strategic planning, often need to take into account large quantities of input information, multiple alternatives, and different objectives. The authors of [25] stated that theoretical approaches to multicriteria decision-making problems have been adapted to practical application through multicriteria decision analysis (MCDA). Common steps in the first stage of MCDM are to:

- Identify objectives, criteria, or topics of relevance to the decision;
- Identify and select stakeholders;
- Identify and/or develop alternatives;
- Weigh criteria and/or define the hierarchy of objectives;
- Iterate if necessary (reviewing all items above).

One of the most accepted MCDM techniques is the Analytical Hierarchy Process (AHP) [26,27]. The goal of any MCDM method is to decide the appropriate weight of each criterion in the model. Paper [28] gives an exhaustive survey of applications of AHP in the field of sustainable development. The main idea behind AHP is to simplify complex decision-making by splitting complex choices into a pairwise comparison of criteria. AHP uses the concept of hierarchy structuring (goal at the highest level, criteria/subcriteria at the intermediate level, and alternatives at the lowest level). At each level, the pairwise comparison represents a PCM (pairwise comparison matrix—judgment matrix). The decision-maker or expert performs weight evaluation for the construction of PCM for comparing a higher-level element to a lower-level element. If, in the end, all elements in the PCM follow the same rule, i.e., we can create a connection between criteria using the transitivity, then the matrix is said to be consistent. By adapting the weights, the aim is to make PCM as close to a consistent matrix as possible. There are multiple consistency indices [29] used to evaluate how "close" the PCM is to a consistent matrix and to help guide weight adjustment. Meta-heuristic optimization techniques have received notable attention among researchers and engineers in the last two decades. Several different, including nature-inspired, techniques have been discussed in detail in [30]. As finding the optimal weights for criteria is an optimization problem, these techniques are also of interest to researchers of MCDM techniques.

In our proposal, previously described I/O multipliers are used as a quantitative measure of the proposed alternatives extracted from the SWOT analysis. As in previously identified steps of MCDM, we will identify objectives and stakeholders and develop alternatives using the results of the SWOT analysis. A key novelty in our proposal is the usage of I/O multipliers as a quantitative measure of the identified alternatives for criteria weighing.

## 3. The Ontology Conceptualization

The goal of the proposed SPO ontology is to ensure a formal, computer-readable representation of the strategic planning process, development priorities, specific goals, and answers to questions such as *what* development priorities are (strategic elements), *how* they will be implemented (action plan elements), *who* is responsible for their implementation

(action plan elements), and *where* and *why* they are being implemented in the first place (strategic elements).

To build the ontology, we used the Methontology methodology [6], and the Protégé editor [31] was used to create machine-readable ontologies. The OWL language [32] was used to formalize the SPO ontology. SPARQL [33], both language and protocol, are used for ontology prompts and Pellet [34] was used for reasoning.

We used the ontology for SP process [7] as the basis for the ontology construction. The semantic representation of the SWOT analysis from [7] as a part of the strategic planning process is extended with the ontology concepts of the regional administrative bodies and development funds given in [35]. To ensure the basis for the ontology-driven simulation method for assessing economic impact, the SPO ontology was annotated with multipliers obtained from the economic impact model described by the Australian Institute for Social Research [24].

The taxonomy of the main concepts of the SPO ontology that semantically represents an SP process and the planning strategy is shown in Figure 1. These concepts are connected to the *is-a* relations.

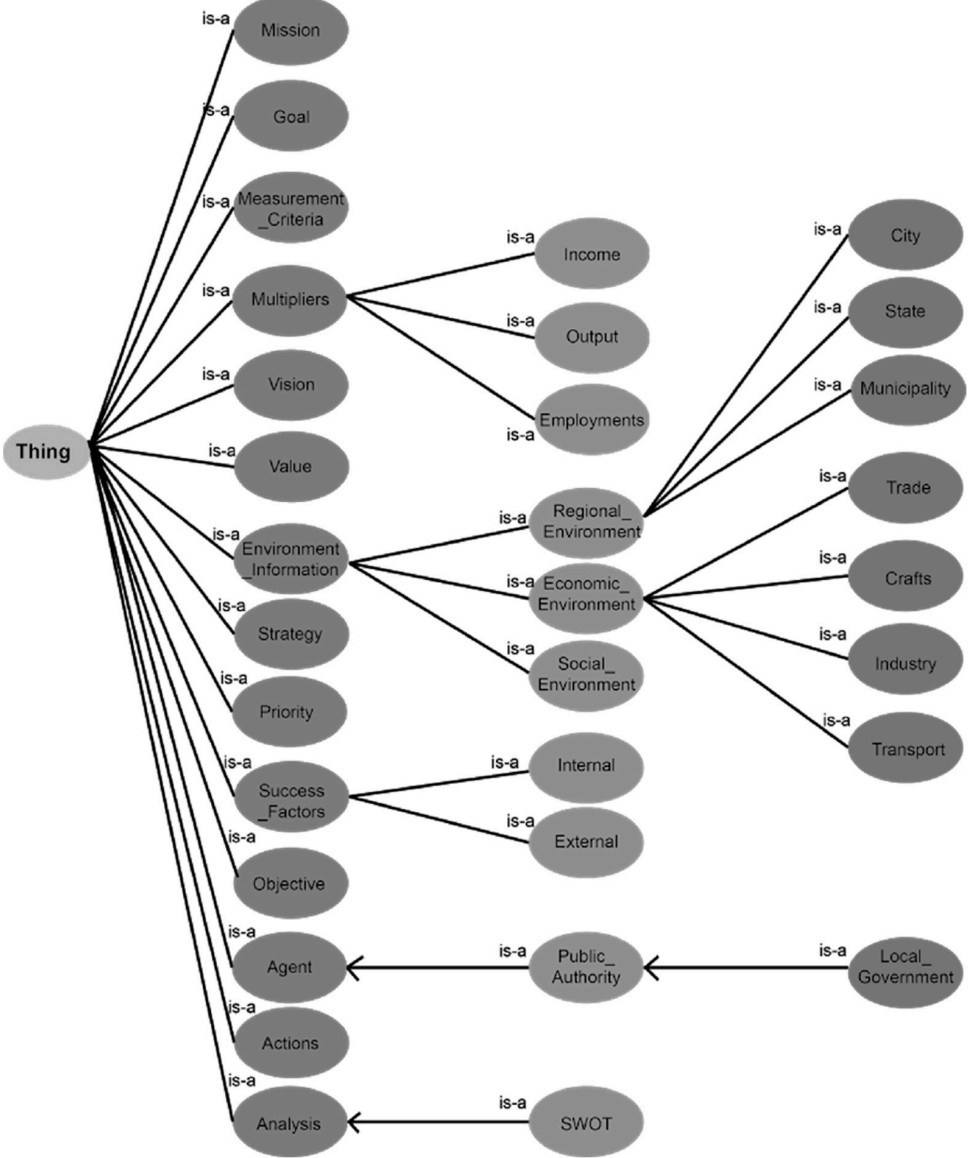

**Figure 1.** SPO ontology.

The ontology proposed by [7] is extended as follows:

To ensure the necessary elements for the impact analysis model construction, geographical, industrial, and business sector data should be provided. To fulfill this, we extend the ontology from [7] with an ontological concept that describes the spatial aspect of the analyzed domain, which includes populated areas and their administrative organization (city, county, and municipality). The concept of *Regional_environment* is illustrated in Figure 2.

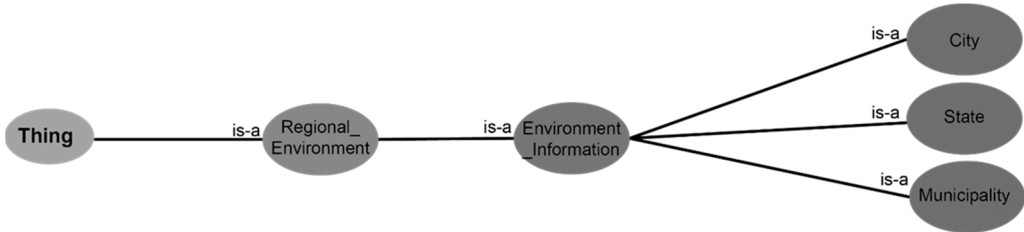

**Figure 2.** *Regional_environment* concept.

As shown in Figure 2, the *Environment_information* concept provides geographical data that will be used by the impact analysis model.

Next, we extend the *Environment_information* concept using the concepts of *Economy_environment* and *Social_environment*. These two concepts describe the socio-economic parameters of the relevant region and business sector data. They provide insight into the current state and enable the identification of problems and limits of development. This extension is illustrated in Figure 3.

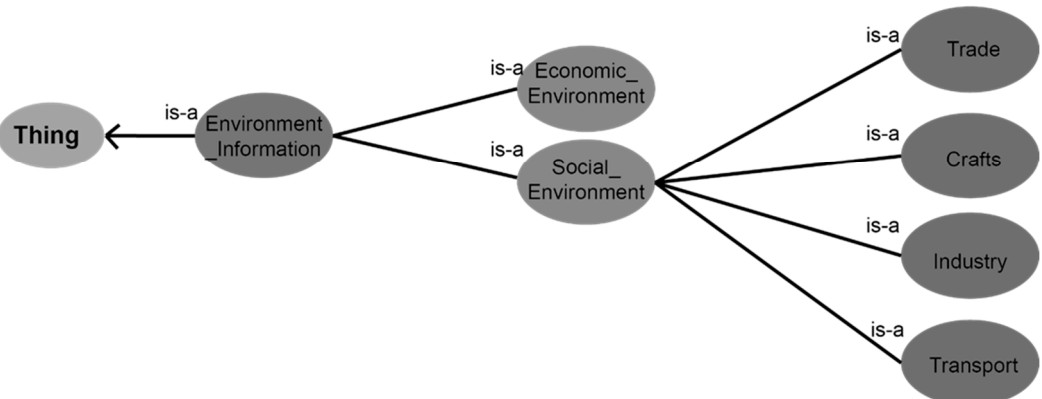

**Figure 3.** *Environment_information* concept.

The *agent* concept from [7] represents the stakeholders involved in the SP process. To ensure the semantic representation of the state and local government, we created the *Public_authority* concept. The *agent* concept from [7] was extended to the created *Public_authority* concept (Figure 4).

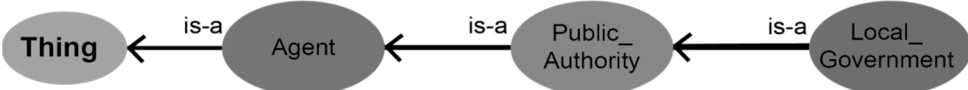

**Figure 4.** *Agent* concept.

The *priority* concept defines the priority levels of the chosen strategic objectives and grades every priority level. In this way, the ontology is enabled to grade the priorities of the objectives. The priorities should be given by professionals after assessing the economic impact of the selected strategic goals and objectives. This ontological concept predicts four priority levels (*LowPriority*, *MiddlePriority*, *HighPriority*, and *TopPriority*), and every level has been assigned a value. The priority assessment and priority index value procedure are

described in detail in the following section. The concept *action* from [7] was extended using the *priority* concept (Figure 5).

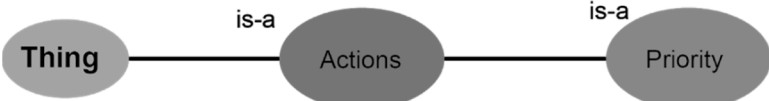

**Figure 5.** *Action* concept.

The ontology in [7] was extended using the *multiplier* concept. The *multiplier* concept represents an I/O multiplier, and its values are obtained from the economic impact analysis of the analyzed region (Figure 6).

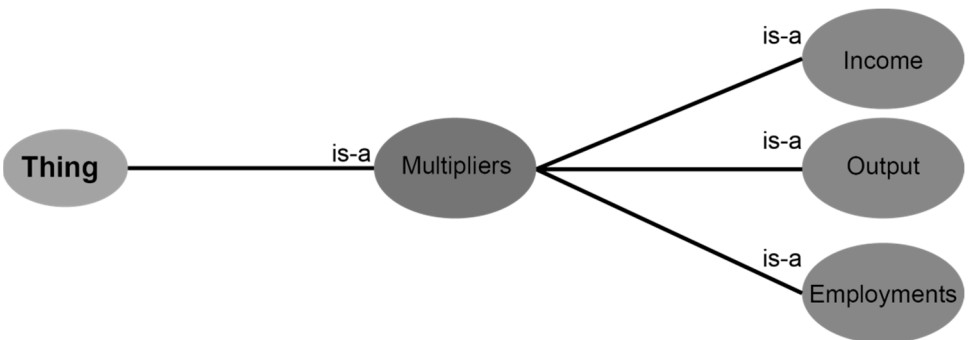

**Figure 6.** *Multiplier* concept.

In Table 1 the object properties and instances of the concepts with which the ontology from [7] is extended are shown.

**Table 1.** Concepts and object properties.

| Concept | Object Properties |
|---|---|
| City | *BelongsToMunicipality*<br>*HaveStrenght*<br>*HaveOpportunity*<br>*HaveStrategy* |
| Municipality | *HaveCity*<br>*BelongToState* |
| State | *HaveMunicipality* |
| Economic_environment | *HaveMultiplyer*<br>*HavePriority* |
| Social_environment | *HaveLow*<br>*HaveHigh* |
| Goal | *HaveObjective* |
| Mission | *DefinedBy* |
| Objective | *PartOfStrategy* |
| Vision | *HaveGoal*<br>*HaveObjective* |

## 4. Ontology Implementation

There are three goals in the process of ontology implementation:

(a) Research-based numerical evaluation of the input–output multipliers and ontology population with them.

(b) Ontology population with results of SWOT analysis.

(c)   Ontology population with instances belonging to *the priority* concept, *regional environment*, and *economic environment* concepts are shown in Figure 1.

Input–output multipliers are indicators that describe the connection between the selected sector and the rest of the regional economy. They can also be used to assess the impact of changes in the selected sector on the rest of the economy [24]. Three types of multipliers can be calculated from an I/O model: output, employment, and income. Multipliers can be divided into three levels: direct, indirect, and induced.

Bekhet [36] considers that the levels of employment in industries related to production, such that the employment/output ratio, can be defined for all levels of output. When applied to the I/O model, it allows the calculation of employment multipliers. The authors suggest that it is possible to estimate the relationship between the output value of the sector and employment in the sector. In this way, we can calculate the income multiplier for each sector. Bekhet used the *outputemployment* relationship to calculate the employment multiplier. According to Bekhet, income multipliers describe the impact of changes in consumer demand, depending on the change in household income (labor force). The input multiplier translates the initial assessment outputs from the valuation results of employment (household income).

To create a basis for the ontology-driven simulation method for assessing economic impact, the SPO ontology is populated with multipliers obtained from the economic impact model created for the analyzed region in the following manner: Every concept that belongs to the set of *Economic_environment* sub-concepts (Figure 1) is populated with instances, object properties, and data properties.

Table 2 presents the *Economic_environment* sub-concept population.

**Table 2.** Concept, object property and sub-concept population for *Economic_environment*.

| Domain Concept | Instance | Object Property | Range Concept | Instance | Data Property |
|---|---|---|---|---|---|
| Industry | Construction_industry | HaveMultiplier | Multipliers | E1 | Value = 11.3 |
| | | | | I1 | Value = 1.3 |
| | | | | O1 | Value = 1.4 |

Instances E1, I1, and O1 represent the multiplier types of Employment, Income, and Output. I/O multipliers for the analyzed region are obtained from the research, output, income, and employment multipliers in the Malaysian economy [36].

The SWOT analysis utilized the data identified during the socio-economic analysis of the analyzed region. The socioeconomic analysis is the analytical basis for starting a strategic planning process. This enables insight into the existing state, identification of problems, and developmental limits. Based on data that have been identified during socio-economic analysis, SWOT analysis matrices are created with entries such as strengths, weaknesses, threats, and opportunities for all key sectors of the analyzed region. The ontology concept *Economic_environment* is populated with SWOT analysis results using the relations *HaveStrenght* or *HaveOpportunity*.

Regarding ontology population with priority values—the *Priority* concept is based on the results obtained and analyzed after assessing the economic impact of the chosen strategy goals using the method proposed in this paper. Once the ontology is formally defined and exactly specified (using the instances), and all information of the analyzed region is stored in the ontology, the SPO ontology can be reused for the creation of subsequent strategies.

## 5. Model Evaluation

The prototype example is presented with the help of a strategic planning case for the city Iskandar Puteri (Iskandar Puteri is a newly developed planned city in Johor Bahru District, Johor, Malaysia), for which the simulation method for assessing the economic impact of the chosen strategy is presented.

Because the goal of this example is to show the use of the proposed method, multiplier values in this example are obtained from [36] and the results of the SWOT analysis were given arbitrarily without prior realization of the implementation described in the lines above. Ontology is populated with a few instances to demonstrate the functionality of the proposed method, as shown in Figure 7.

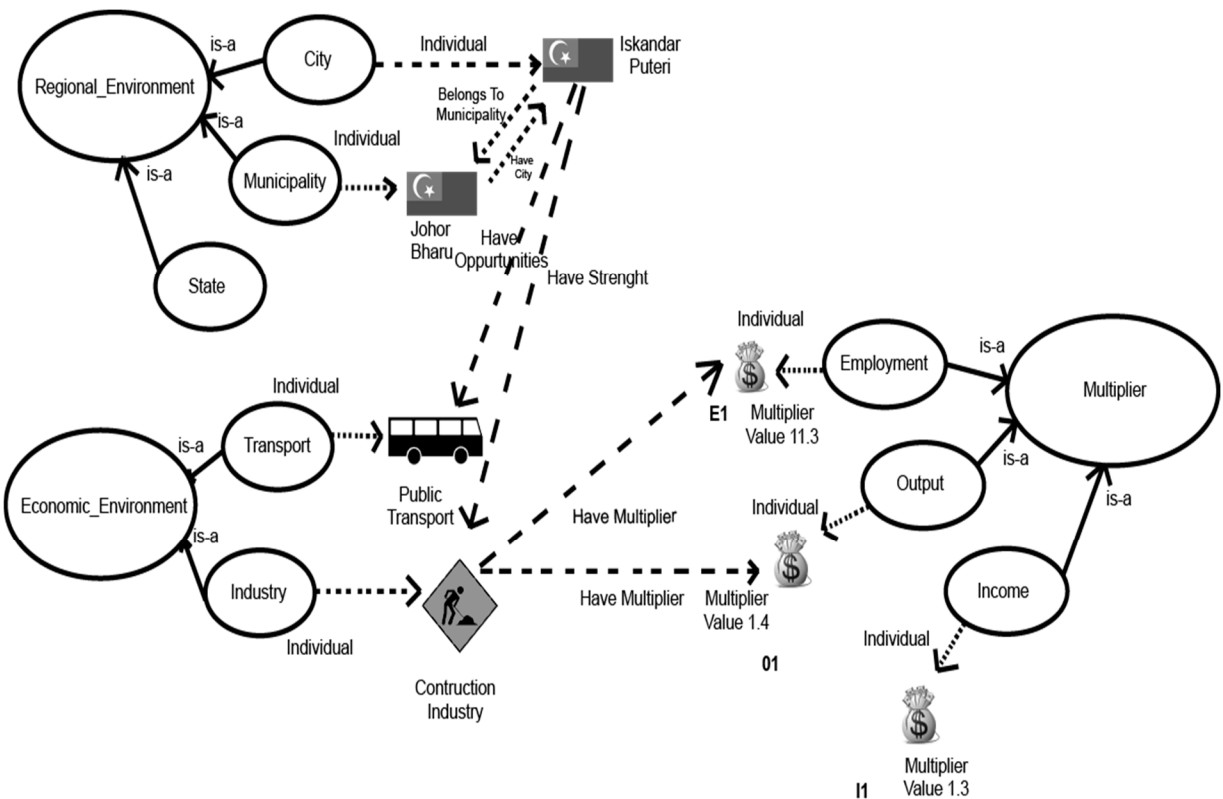

**Figure 7.** Populated ontology.

Tasks of the strategic planning case for the city Iskandar Puteri are:

(1)  To detect possible strategic goals according to SWOT analysis results.
(2)  To simulate the economic impact of chosen strategic goals, with a budget of 3 million MYR for increasing final demand.
(3)  To choose the best strategy goals.

### 5.1. Detection of Possible Strategic Goals According to SWOT Analysis Results

To choose the possible strategic goals, we first need to obtain the SWOT analysis results [37]. The results are shown in Figure 7. Ontology is populated with the results of the SWOT analysis. We queried the ontology using the Controlled Natural Language Protégé plugin ACE View [38]. ACE View is an ontology and rule editor that uses Attempto Controlled English (ACE) to create, view, edit, and query ontology web language (OWL) ontologies and semantic web rule language (SWRL) rules.

The results obtained from ACE View word usage in SPO ontology for variable Iskandar Puter are: *Johor_Bahru Have_cities Iskandar_Puteri*, *Iskandar_Puteri is a city*, *Iskandar_Puteri Have_strenghts Construction_industry*, *Iskandar_Puteri Have_opportunities Public_transport*, *Iskadar_Puteri HaveLow Employment_rate*.

The results show that the city of Iskandar Puteri has the opportunity to develop public transport and has strength in the construction industry. In this case, we select the construction industry as a possible strategic goal.

### 5.2. Simulation of the Economic Impact of Chosen Possible Strategic Goals with a Budget of 3 Million MYR for Increasing Final Demand

To simulate the economic impact of the identified possible strategic goals (we chose the construction industry), we used I/O multipliers from the ontology shown in Figure 7 and the ontology plugin ACE View. In this phase, we query ontology using ACE View to obtain multiplier values and calculate the economic impact for the possible strategic goal chosen in the previous step of the construction industry. The employment multiplier E1 (Figure 7) gives the number of direct jobs related to the specific industry and indirect and induced jobs in the support industry, and output multiplier O1 (Figure 7) provides direct and indirect output impacts on the industry.

The results obtained from ACE View word usage in the SPO ontology for variable construction_industry are: *Construction_industry haveMultipliers O1*, *Construction_industry haveMultipliers E1*, *O1 is an Output*, *O1 is 1.4*, *E1 is Employment*, and *E1 is 11.3*.

Multiplying 3 million (defined in the strategic planning case) with a multiplier value obtained from an ontology, we can conclude that the total direct and indirect output impacts on industry of region will be $3 \times 1.4 = 4.2$ mil. MYR. *Output indicator value = 4.2 mil*.

Multiplying 3 million (increase in final demand) with a given employment multiplier, we can conclude that the total direct and indirect impacts on employment changes resulting from this additional output in the construction industry will be $3 \times 11.3 = 33.9$, and approximately 34 new jobs will be created from this investment. *The employment indicator value is 34*.

### 5.3. Choosing the Best Strategic Goals

We repeat the process explained in (B) for the public transport sector and compare the values of the calculated indicators with the values obtained for the construction industry. After analysis, we will have a basis to decide how to allocate the budget at its disposal and achieve the best effects for the economy of the region.

### 5.4. Discussion

The authors of [39] stated that an ontology model can effectively be used as a systematic procedure for hierarchy analysis in an analytical system environment. Similar to the proposed *Economic_envirnoment* concept in our paper, the authors of [40] define domain ontology characteristics of alternatives.

Application of ontological descriptions in the decision-making process can successfully validate chosen alternatives. Utilizing the I/O multipliers in our ontology we introduced a quantitative measure of the proposed alternatives.

SWOT analysis results were used to identify proposed alternatives.

I/O multipliers play a crucial role in determining the validity of a decision, as they are used for alternative weighing.

This ontology is developed to support decision-making for the strategic planning in local/state government institutions. However, the main concepts of the ontology, and usage of input/output multipliers as a quantitative measure to support arguing *why* some strategic goal should be prioritized, can be adopted to cover other fields. For industrial strategic planning, the concept of *Regional_Environment* would need to be adapted to some concept that better fit the specific industry use case, and *Economic_Environment* could also be expanded. Although SWOT analysis would be also adapted, the basic concept of using SWOT analysis results to populate proposed alternatives would still be viable.

## 6. Conclusions

In this study, we present a novel approach to defining strategic goals and objectives. The presented approach is based on the need to integrate business intelligence and decision support tools into a strategic planning process. For this purpose, the authors propose an ontology-driven simulation method for assessing the economic impacts that should be implemented as decision support tools for choosing the best strategy goals and objec-

tives. The proposed method ensures the automatization of the decision-making process in the strategic planning process. The authors demonstrated the implementation of the proposed method by analyzing the results of the presented strategic planning case. An ontology-driven simulation method for assessing economic impacts ensures the possibility of estimating the full income and job effects of changes in business activity levels, provides support for making strategic decisions, and provides an answer to the question of how to allocate the budget at its disposal and achieve the best effects for the economy of the region during the strategic planning process.

The novelty of the research presented in this paper is reflected by the development of an ontological model of actions and reasons, that is, semantically described combinations of activities and requirements that are defined by development strategies. The proposed approach for creating an SPO ontology ensures a formalized, computer-readable description of the elements of the strategic planning process. The proposed methodological approach creates a common dictionary of terms used by all participants in the strategy development process.

The ontological representation of development strategies (the process of creating development strategies is specified by a group of concepts and concept instances) creates conditions for the reuse of semantically described strategies in the process of creating new strategies for the relevant domain. Instances that populate the ontology in the strategy development process are machine-readable and may be used in the creation of a new strategy or redesign of a current strategy.

The creation of the described ontology is the first step in building an automatized model of development strategy creation. The semantic representation of SWOT analysis enables the creation of tools that simplify the process of strategy and action plan generation. The use of the SPO ontology in an ontologically-driven information system enables automation and unification of development strategies and action plan creation in the framework of designing strategic plans for the development of the analyzed region.

**Author Contributions:** Conceptualization: M.Z., S.A., B.M., P.V. and V.P.; methodology: S.A., B.M. and V.P.; writing—original draft preparation: S.A., B.M., M.Z., P.V. and V.P.; writing—review and editing: M.Z., S.A., B.M., P.V. and V.P.; supervision, B.M. and S.A. All authors have read and agreed to the published version of the manuscript.

**Funding:** This research received no external funding.

**Institutional Review Board Statement:** Not applicable.

**Informed Consent Statement:** Not applicable.

**Data Availability Statement:** Data (OWL files) available in a publicly accessible repository that does not issue DOIs. This data can be found here: https://drive.google.com/drive/folders/1u-oIzKsPetTPBAWb-dtsYUna4BhPZZso.

**Conflicts of Interest:** The authors declare no conflict of interest.

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
