# Peer review of "An Approach to the Semantic Representation of the Local Government Strategic Planning Process: Ontology-Driven Simulation Method for Assessing Economic Impacts"

_applsci, doi:10.3390/app13031258_

Round 1
Reviewer 1 Report
It is not clear how the proposed ontology helps to solve the problems of strategic planning. If strategic planning is related to the questions that are named in lines 80-87, then how does the proposed model answer these questions? There are no examples of SPARQL requests, algorithms. It is not clear how the SWOT analysis is related to this ontology. It is not clear how the ontology is related to the equation 1.
I am not sure that the ontology in Figure 1 is complete and sufficient for strategic planning tasks
There is no discussion section where you can compare the results with those of other authors.
I do not see the scientific contribution of this article.
It seems to me that the article is not suitable for an international scientific journal.
Author Response
We, the authors, would like to thank the reviewer for her/his valuable input and comments. We would like to answer some of the concerns/questions stated in the review:
Q1: It is not clear how the proposed ontology helps to solve the problems of strategic planning. If strategic planning is related to the questions that are named in lines 80-87, then how does the proposed model answer these questions?
Answer: The vision of the strategy created as defined at lines 80-87 does not answer the question of why we select specific strategic goals. The adoption of vision statements and the definition of the objectives of the strategy is a decision-making process. The ontology proposed in this paper will give chance to strategy creators to evaluate selection of the strategic goals.
Q2: There are no examples of SPARQL requests, algorithms
Answer: Distinguished reviewer, we were using rule editor as stated in lines 503-510 (of the updated manuscript), ACE View is an ontology and rule editor that uses Attempto Controlled English (ACE) to create, view, edit, and query ontology web language (OWL) ontologies and semantic web rule language (SWRL) rules.
The results obtained from ACE View word usage in SPO ontology for variable Iskandar Puteri are: Johor_Bahru Have_cities Iskandar_Puteri, Iskandar_Puteri is a city, Iskandar_Puteri Have_strenghts Construction_industry, Iskandar_Puteri Have_opportunities Public_transport, Iskadar_Puteri HaveLow Employment_rate.
Q3: It is not clear how the ontology is related to equation 1.
Answer: Distinguished reviewer , equation 1. represents input-output model consumption matrix. The input-output model is a quantitative economic technique that represents the interdependencies between different branches of the national economy or different regional economies. In proposed ontology we used multipliers generated by the regional input-output model to answer the question: why specific strategic goals are selected. The revised manuscript contains new passages (currently in red for easier distinction) that now, we hope, better explain how input-outpu model is realted to ontology.
Q4: It is not clear how the SWOT analysis is related to this ontology
Answer: Distinguished reviewer, as such, development strategies should stimulate sectors with growth potential for the economy and employment. To populate designed ontology we used SWOT analysis to collect information. The results of SWOT analysis were then used to populate the ontology concept Economic_environment using the relations HaveStrenght or HaveOpportunity. This is now also emphasized in text of the manuscript.
Q5: I am not sure that the ontology in Figure 1 is complete and sufficient for strategic planning tasks.
Answer: Distinguished reviewer, Figure 1. shows only main concepts of the ontology without additional designed relations to make model simpler and more readable.
Q6: There is no discussion section where you can compare the results with those of other authors.
Answer: In the related work, we comment on the results from other authors, but, to our best knowledge there is no similar approach to compare more directly.In accordance with Your and another reviewer requests manuscript has been updated to address stated concerns. We hope You will find updated version improved compared to original submission

Reviewer 2 Report
In this manuscript, the authors propose an Ontology-driven simulation method for the creation of a strategic planning ontology for local governments. However, I have few observations:
1. Need to handle a large volume of diverse information and the occurrence of a great number of alternatives and multi-objective criteria when choosing them are the reason behind development of various MCDM techniques and Ontology-driven models. Authors are required to elaborate more clearly on the purpose and novelty of their work and add the future scope in the conclusion section.
2. Please add a subsection clearly articulating the main limitations and wider applicability of your method.
3. Used of various MCDM techniques like AHP (Analytical hierarchy process) and metaheuristics are quite common in modern-day decision science problems having hierarchy analysis issues. Authors are required to go through recent references related to various recently developed in MCDM and metaheuristics to make the reference list exhaustive. For example:
(i) Nesterenko, O. (2020, May). Ontology and Analytic Hierarchy Process in the information and analytical systems. In International Scientific Conference “Intellectual Systems of Decision Making and Problem of Computational Intelligence” (pp. 302-314). Springer, Cham.
(ii) Ram, M., et al. Meta-heuristic Optimization Techniques: Applications in Engineering, Berlin, Boston: De Gruyter, 2022. https://doi.org/10.1515/9783110716214.
(iii) Rawat, S., et al. (2022). A State-of-the-Art Survey on Analytical Hierarchy Process Applications in Sustainable Development. International Journal of Mathematical, Engineering and Management Sciences, 7(6), 883-917. https://doi.org/10.33889/IJMEMS.2022.7.6.056.
(iv) Pant, S., et al. (2022). Consistency Indices in Analytic Hierarchy Process: A Review. Mathematics, 10(8), 1206.
(v) Kumar, A., & Pant, S. (2022). Analytical hierarchy process for sustainable agriculture: An overview. MethodsX, 101954.
4. Recheck all the captions of figures, equations and tables.
5. However, the English of the manuscript is readable though I suggest proofreading of the manuscript carefully for grammatical errors.
Thanks.
Author Response
We would like to thank You for your valuable input and comments on our manuscript. We would like to respond to your questions/raised issues.
Issue 1. Need to handle a large volume of diverse information and the occurrence of a great number of alternatives and multi-objective criteria when choosing them are the reason behind development of various MCDM techniques and Ontology-driven models. Authors are required to elaborate more clearly on the purpose and novelty of their work and add the future scope in the conclusion section.
Distinguished reviewer, thank you for this constructive comments. The proposed approach in our paper can be categorized as MCDM quantitative criteria weighing technique. According to provided suggestions, we expanded the text of the manuscript to correlate our proposal to other techniques.
Issue 3. Used of various MCDM techniques like AHP (Analytical hierarchy process) and metaheuristics are quite common in modern-day decision science problems having hierarchy analysis issues. Authors are required to go through recent references related to various recently developed in MCDM and metaheuristics to make the reference list exhaustive.
With regards to issues 1 and 3 we extended the manuscript with following:(extensions are in red in manuscript to make it easier to distinguish it from original submission)
In the introduction section
The discussion above makes it clear that strategic planning is a decision-making process, based on multiple, and sometimes opposing criteria. Multicriteria decision-making (MCDM) techniques assist decision-makers in dealing with large amounts of complex information. Some approaches to MCDM are discussed in brief in the next section in correlation with the proposed ontology.
In the Related works section
As well as ontology-driven models, for decision-making problems, different MCDM techniques have been study topics since the 1960s. All decision-making problems, as is the case with strategic planning, often need to take into account large quantities of input information, multiple alternatives, and different objectives. Authors in [25] stated that theoretical approaches to multicriteria decision-making problems have been adapted to practical application through multicriteria decision analysis (MCDA). Common steps in the first stage of MCDM are to:
- identify objectives, criteria, or topics of relevance to the decision
- identify and select stakeholders
- identify and/or develop alternatives
- weigh criteria and/or define the hierarchy of objectives
- iterate if necessary (reviewing all items above)
One of the most accepted MCDM techniques is Analytical Hierarchy Process (AHP) [26, 27]. The goal of any MCDM method is to decide the appropriate weight of each criterion in the model. Paper [28] gives an exhaustive survey of applications of AHP in the field of sustainable development. The main idea behind AHP is to simplify complex decision-making by splitting complex choices into a pairwise comparison of criteria. AHP uses the concept of hierarchy structuring (goal at the highest level, criteria/subcriteria at the intermediate level, and alternatives at the lowest level). At each level, the pairwise comparison represents a PCM (pairwise comparison matrix - judgment matrix). The decision-maker or expert performs weight evaluation for the construction of PCM for comparing a higher-level element to a lower-level element. If, in the end, all elements in the PCM follow the same rule, ie. we can create a connection between criteria using the transitivity then the matrix is said to be consistent. By adapting the weights, the aim is to make PCM as close to a consistent matrix as possible. There are multiple consistency indices [29] used to evaluate how “close” the PCM is to a consistent matrix and to help guide weight adjustment. Meta-heuristic optimization techniques have received notable attention among researchers and engineers in the last two decades. Several different, including nature-inspired techniques, have been discussed in detail in [30]. As finding the optimal weights for criteria is an optimization problem, these techniques are also of interest to researchers of MCDM techniques.
In our proposal, previously described I/O multipliers are used as a quantitative measure of the proposed alternatives extracted from the SWOT analysis. As in previously identified steps of MCDM, we will identify objectives, stakeholders and develop alternatives using the results of the SWOT analysis. A key novelty in our proposal is the usage of I/O multipliers as quantitative measure of the identified alternatives, for criteria weighing.
Issue 2. Please add a subsection clearly articulating the main limitations and wider applicability of your method.
In the discussion of the proposed model we added:
Authors in [39], stated that an ontology model can effectively be used as a systematic procedure for hierarchy analysis in an analytical system environment. Similar to the proposed Economic_envirnoment concept in our paper, authors in [39] define domain ontology characteristics of alternatives.
Application of ontological descriptions in the decision-making process can successfully validate chosen alternatives. Utilizing the I/O multipliers in our ontology we introduced a quantitative measure of the proposed alternatives.
SWOT analysis results were used to identify proposed alternatives.
I/O multipliers play a crucial role in determining the validity of a decision, as they are used for alternative weighing.
This ontology is developed to support decision-making for the strategic planning in local/state government institution. However, the main concepts of the ontology, and usage of input/output multipliers as quantitative measure to support arguing why some strategic goal should be prioritized can be adopted to cover other fields. For industrial strategic planning the concept of Regional_Environment would need to be adapted to some concept that better fit the specific industry use case, and Economic_Environment could also be expanded. Although SWOT analysis would be also adapted, the basic concept of using SWOT analysis results to populate proposed alternatives would still be viable. Several new refenerces are added in the process:[25] G. Kiker, T. Bridges, A. Varghes, T. Seager and I. Linkov, Application of Multi-criteria decision analysis in environmental decision making, vol. 1(2), 2005, pp. 95-108.
[26] T. Saaty, Multicriteria Decision Making: The Analytic Hierarchy Process, New York: McGraw-Hill, 1980.
[27] T. Saaty, Fundamental of Decision Making and Priority Theory with the AHP, Pittsburgh: RWS Publications, 1994.
[28] S. S. Rawat, S. Pant, A. Kumar, M. Ram, H. K. Sharma and A. Kumar, "A state-of-the-art Survey on Analytical Hierarchy Process Applications in Sustainable Development," International Journal of Mathematics, Engineering and Management Services, vol. 7, no. 6, pp. 883-917, 2022.
[29] S. Pant, A. Kumar, M. Ram, Y. Klochkov and H. K. Sharma, "Consistency Indices in Analytic Hierarchy Process: A Review," Mathematics, vol. 10, no. 1206, 2022.
[30] M. Ram, A. Kumar, S. Pant and O. Yadav, Eds."Meta-heuristic Optimization Techniques," De Gruyter Series on the Applications of Mathematics in Engineering and Information Sciences, 2022.
[39] Nesterenko, O. (2021). Ontology and Analytic Hierarchy Process in the Information and Analytical Systems. In: Babichev, S., Lytvynenko, V., Wójcik, W., Vyshemyrskaya, S. (eds) Lecture Notes in Computational Intelligence and Decision Making. ISDMCI 2020. Advances in Intelligent Systems and Computing, vol 1246. Springer, Cham. https://doi.org/10.1007/978-3-030-54215-3_19
Cross referencing the literature from the manuscript text as well as Figures and Tables are updated so it should be ok.
We did one more round of english proof reading to enhance the language of the manusctipt.
We hope that you will find updated manuscript as an improvement over the original submission.
Best regards, authors
